# DomainStudio: Fine-Tuning Diffusion Models for Domain-Driven Image Generation using Limited Data

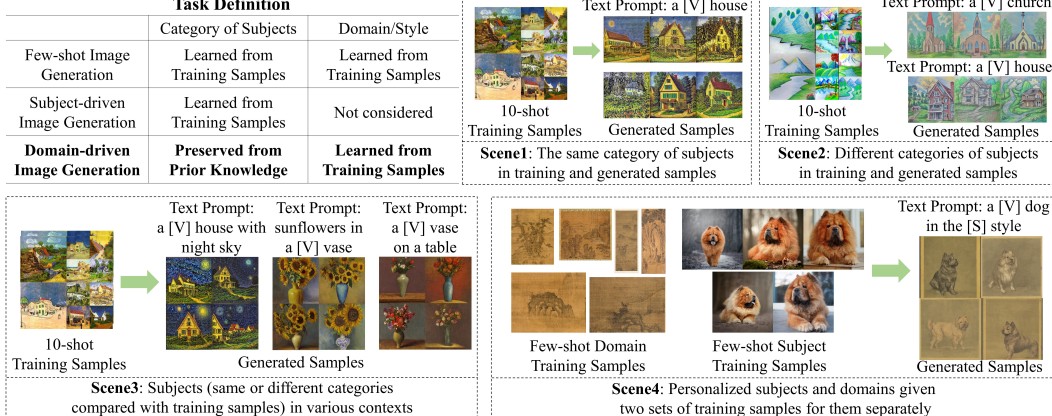

Figure 1: We introduce a DDPM-based AI image creation approach named DomainStudio for domain-driven image generation, which preserves subjects provided by prior knowledge and adapts them to the domains (e.g., styles) learned from training samples. It tackles a task different from few-shot or subject-driven generation. The key difference is that the category of subjects in the generated samples is preserved from prior knowledge (e.g., pre-trained models) and can be different from the subjects in training samples in domain-driven generation. DomainStudio is compatible with both unconditional and text-to-image generation and only needs few-shot training data, extending DDPMs to produce compelling results in various application scenarios as shown above.

## Abstract

Denoising diffusion probabilistic models (DDPMs) have been proven capable of synthesizing high-quality images with remarkable diversity when trained on large amounts of data. Unfortunately, they are still vulnerable to overfitting when fine-tuned on extremely limited data. Existing works have explored subject-driven generation with text-to-image (T2I) models using a few samples. However, there still lack effective data-efficient methods to synthesize images in specific domains (e.g., styles), which remains challenging due to ambiguities inherent in natural language and out-of-distribution effects. This paper introduces a few-shot fine-tuning approach named DomainStudio to realize domain-driven image generation, which is defined as retaining the subjects from prior knowledge provided by pre-trained models and adapting them to the domain extracted from training data, pursuing high quality and great diversity like prior few-shot generation methods. We propose to keep the image-level relative distances between adapted samples and enhance the learning of high-frequency details from both pre-trained models and training samples. DomainStudio is compatible with both unconditional and T2I DDPMs. This work makes the first attempt to achieve unconditional DDPM-based image generation using limited data, achieving better visual effects than current GAN-based approaches under the same task settings. For T2I generation, DomainStudio is qualified for synthesizing samples in domains characterized by few-shot training data. It extends the applicable scenarios of modern large-scale T2I diffusion models, which are insufficient to be handled with existing few-shot fine-tuning methods of T2I DDPMs like Textual Inversion and DreamBooth.

# 1 INTRODUCTION

Deep generative models (Goodfellow et al., 2014; Karras et al., 2020b; Kingma & Welling, 2013; Ho et al., 2020) have made significant progress in synthesizing high-quality and diverse images over past years. DDPMs (Sohl-Dickstein et al., 2015) have become the most prevailing approach amongst all of them with their outstanding performance and usability under various conditions (Nichol & Dhariwal, 2021; Li et al., 2023; Zhang & Agrawala, 2023; Rombach et al., 2022).

Despite the remarkable success, DDPMs are heavily dependent on large-scale training data (Schuhmann et al., 2022; Van den Oord et al., 2016; Yu et al., 2015) and vulnerable to overfitting when trained on limited data (Moon et al., 2022) like other generative models. A series of methods have been proposed (Wang et al., 2018; Karras et al., 2020a; Mo et al., 2020; Wang et al., 2020; Li et al., 2020; Ojha et al., 2021; Zhao et al., 2022b;a; 2023) to address such problems for GANs by transferring knowledge from pre-trained models to target domains. However, they are completely unconditional and still get limited fidelity influenced by unexpected blurs and deformations.

Existing data-efficient methods based on DDPMs like LoRA (Hu et al., 2021), Textual Inversion (Gal et al., 2022), and DreamBooth (Ruiz et al., 2023) mainly focus on subject-driven generation which aims to preserve the key features of customized subjects and synthesize novel scenes, poses, and views. It remains challenging to synthesize samples in specific domains (e.g., styles) even with powerful large-scale T2I diffusion models (Saharia et al., 2022; Ramesh et al., 2022; Rombach et al., 2022). Firstly, it is difficult to describe domains with text prompts accurately for T2I models due to the ambiguities in natural language. Besides, even if pre-trained models have captured typical domains owing to the presence of corresponding samples in their training data, they may still produce biased results influenced by the out-of-distribution effects. For example, there exist different styles in Van Gogh's paintings. Given a text prompt of "Van Gogh style", T2I models may choose a certain style randomly or mix some of them, leading to unsatisfactory outputs.

To this end, we introduce DomainStudio to enable domain learning of both unconditional and T2I DDPMs through few-shot fine-tuning with extremely limited data (e.g., 10 images). We propose to preserve the subjects from pre-trained source models, adapt them to the domain extracted from training samples, and define this task as domain-driven generation. It differs from few-shot image generation (Wang et al., 2018), which learns subject and domain knowledge from training samples, and subject-driven generation, which learns customized subjects and ignores domain knowledge. It is worth noting that "the preserved subjects" in domain-driven generation refer to a category of subjects including diverse individualities (e.g., a dog, a house) instead of specific subjects learned in subject-driven generation. We first propose an image-level pairwise similarity loss for DDPMs to keep the relative distances between adapted samples similar to source samples for the preservation of subject distributions and greater diversity. Then we design a high-frequency details enhancement method from two perspectives, including preserving more details provided by source models and learning more details from limited data for finer quality. As illustrated in Fig. 1, DomainStudio is qualified for a series of domain-driven tasks. Our main contributions are concluded as follows:

- We propose to synthesize samples in specific domains characterized by few-shot data by preserving subjects from source models and learning domain knowledge from training samples.

- We design a domain-driven DomainStudio approach compatible with both unconditional and T2I DDPMs to maintain the distributions of subjects and enhance high-frequency details learning.

- The effectiveness of DomainStudio is demonstrated qualitatively and quantitatively on a series of few-shot domain-driven tasks. For unconditional generation, DomainStudio achieves better diversity and visual effects than current state-of-the-art GAN-based approaches. For T2I generation, DomainStudio achieves compelling results in several scenes which existing few-shot fine-tuning methods of T2I diffusion models cannot handle.

# 2 RELATED WORK

**DDPMs** (Sohl-Dickstein et al., 2015) define a forward noising (diffusion) process adding Gaussian noises $\epsilon$ to training samples $x_0$ and employ a UNet-based neural network $\epsilon_\theta$ to approximate the reverse distribution, which can be trained to predict the added noises or the denoised images. Ho et al. (2020) demonstrates that predicting $\epsilon$ performs well and achieves high-quality results for

unconditional image generation using a reweighted loss function:

$$\mathcal{L}_{simple}^{unc} = E_{t,x_0,\epsilon} \left[ ||\epsilon - \epsilon_\theta(x_t, t)|| \right]^2,$$

(1)

where t and $x_t$ represent the diffusion step and corresponding noised image. DDPMs have achieved competitive unconditional generation results on typical large-scale datasets (Krizhevsky et al., 2009; Yu et al., 2015; Van den Oord et al., 2016). Besides, classifier guidance is added to realize DDPM-based conditional image generation (Dhariwal & Nichol, 2021). Latent diffusion models (Rombach et al., 2022) employ pre-trained autoencoders to compress images into the latent space and achieve high-quality conditional generation using inputs such as text, images, and semantic maps.

**T2I Generation** (Crowson et al., 2022; Ding et al., 2021; Gafni et al., 2022; Jain et al., 2022; Hinz et al., 2020; Li et al., 2019b;a; Qiao et al., 2019a;b; Ramesh et al., 2021; Tao et al., 2020; Zhang et al., 2018b) has achieved great success based on GANs (Brock et al., 2019; Karras et al., 2019; 2020b; 2021), transformers (Vaswani et al., 2017), and diffusion models (Ho et al., 2020) with the help of image-text representations like CLIP (Radford et al., 2021). Large-scale T2I generative models including Imagen (Saharia et al., 2022), DALL-E2 (Ramesh et al., 2022), and Stable Diffusion (Rombach et al., 2022) further expand application scenarios and improve generation quality. Subject-driven fine-tuning methods like Textual Inversion (Gal et al., 2022) and DreamBooth (Ruiz et al., 2023) realize the personalization of T2I diffusion models using limited data. Custom Diffusion (Kumari et al., 2023) and MixofShow (Gu et al., 2023) explore multi-concept customization of T2I diffusion models. These subject-driven methods ignore the learning of domains. Our work makes up for gaps by tackling the domain-driven task based on DDPMs. Contemporary works (Sohn et al., 2023b;a) based on MaskGIT (Chang et al., 2022) and MUSE (Chang et al., 2023) tackle similar tasks of generating images containing different subjects and learning the style from training samples. Our approach is compatible with both unconditional and T2I DDPMs for domain-driven generation.

**Few-shot Image Generation** aims to achieve high-quality generation with great diversity using limited data. Existing approaches are GAN-based and unconditional. Most works follow TGAN (Wang et al., 2018) to adapt GANs pre-trained on large source datasets to target domains. Following methods can be roughly divided into data augmentation approaches (Tran et al., 2021; Zhao et al., 2020a;b; Karras et al., 2020a), trainable parameters fixing (Noguchi & Harada, 2019; Mo et al., 2020; Wang et al., 2020), and model regularization (Li et al., 2020; Ojha et al., 2021; Zhao et al., 2022b; Zhu et al., 2022; Xiao et al., 2022). AdAM (Zhao et al., 2022a) and RICK (Zhao et al., 2023) explore knowledge transfer between source/training datasets with large gaps. The proposed Domain-Studio follows similar strategies to adapt pre-trained source models to target domains but keeps subjects in source samples. It is compatible with both unconditional and T2I DDPMs. Suppose training samples share the same category of subjects with source samples produced by pre-trained models, the domain-driven DomainStudio shares the same target with few-shot image generation. Otherwise, DomainStudio preserves subjects in source samples and learns domain knowledge.

**High-frequency Components (HFC) in GANs** have been proven to bias the generation quality (Schwarz et al., 2021). Prior GAN-based methods add skip connections of the HFC in features of generators (Yang et al., 2022; Wang et al., 2022), employ additional discriminators for HFC (Wang et al., 2022; Huang et al., 2022), and build HFC alignment between features in generators and discriminators (Wang et al., 2022) to improve generation quality given limited data. Our approach maintains the distributions of the relative distances between image-level HFC in source samples and learns more HFC from limited data, aiming for finer quality in few-shot fine-tuning of DDPMs.

**Neural Style Transfer** (NST) (Gatys et al., 2015; Ghiasi et al., 2017; Deng et al., 2022; Park & Kim, 2022; Tumanyan et al., 2022) transfers an input image to the style prompted by another image while maintaining the input contents. NST methods are designed to disentangle content and style information. Although NST and our work synthesize images in specific styles, they have different targets and cannot be compared directly. NST is image-level while our approach is model-level and aims to obtain adapted models synthesizing diverse samples in target domains.

## 3  METHOD

This section introduces the DomainStudio approach in detail. We propose to preserve the distributions of subjects during domain adaptation by keeping the image-level relative pairwise distances between adapted samples similar to source samples (Sec 3.1). Besides, we guide adapted models

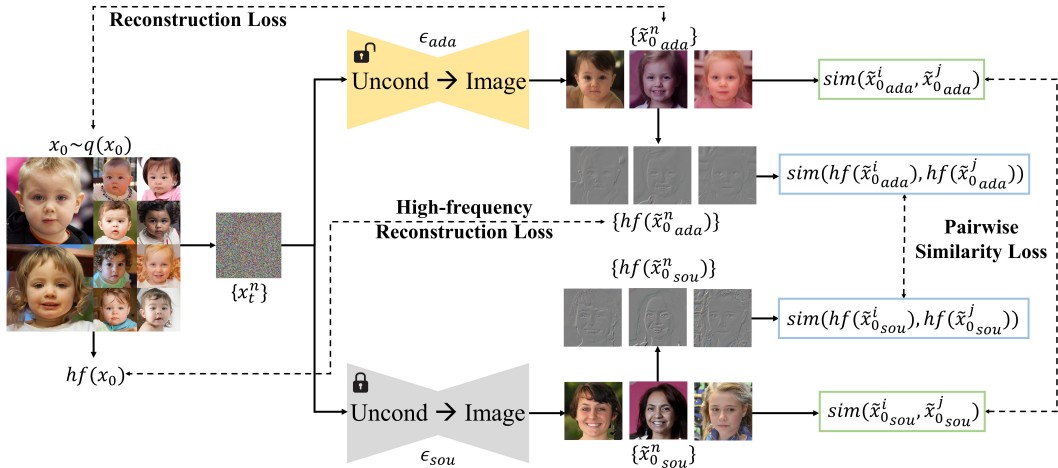

Figure 2: **Overview of DomainStudio on unconditional DDPMs.** We introduce an image-level pairwise similarity loss to maintain the diverse distributions of subjects and guide adapted models to learn knowledge of target domains. We also apply it to high-frequency components for better preservation of high-frequency details and guide adapted models to learn more high-frequency details from limited data with the reconstruction loss between high-frequency components extracted from few-shot data and adapted samples.

to learn more high-frequency details from limited training data and source samples (Sec 3.2). Our approach fixes source DDPMs $\epsilon_{sou}$ as reference for adapted DDPMs $\epsilon_{ada}$. The pre-trained autoencoders ($\mathcal{E} + D$) and text encoders $\Gamma$ used in T2I generation are fixed as well. Adapted models are initialized to source models and adapted to target domains. Overviews of DomainStudio for unconditional and T2I generation are illustrated in Fig. 2 and 3, respectively.

For T2I generation, we define source and adapted samples using text prompts $P_{sou}$ and $P_{tar}$. To avoid using the prior knowledge of target domains provided by the large T2I model, we employ a unique identifier [V] to represent target domains. For example, we define the source and adapted samples with text prompts "a house" and "a [V] house" in Fig. 3. Subjects in source and adapted samples share the same category but can be different from subjects in training samples. The source and target text prompts $P_{sou}$ and $P_{tar}$ are encoded by pre-trained text encoder $\Gamma$ to conditioning vectors $c_{sou}$ and $c_{tar}$. The adapted models are guided to learn from training samples with the reconstruction loss:

$$\mathcal{L}_{simple}^{text} = \mathbb{E}_{t,z_t,c_{tar},\epsilon}||\epsilon_{ada}(z_t, t, c_{tar}) - \epsilon||^2, \tag{2}$$

where $z_t$ represents the latent codes $z$ compressed from training samples added with noises.

### 3.1 SUBJECT DISTRIBUTIONS PRESERVATION

We design an image-level pairwise similarity loss to maintain the relative pairwise distances between adapted samples for subject distributions preservation during domain adaptation. To construct N-way probability distributions for each sample in unconditional image generation, we sample a batch of noised images $\{x_t^n\}_{n=0}^N$ by randomly adding Gaussian noises to training samples $x_0 \sim q(x_0)$ following $x_t = \sqrt{\overline{\alpha}_t}x_0 + \sqrt{1-\overline{\alpha}_t}\epsilon$, where $\overline{\alpha}_t := \prod_{s=0}^t (1-\beta_s)$ and $\beta_s \in (0,1)$ represents the variance at diffusion step $t$. Then both source and adapted models are applied to predict the fully denoised images $\{\tilde{x}_0^n\}_{n=0}^N$. We have the prediction of $\tilde{x}_0$ in terms of $x_t$ and $\epsilon_\theta(x_t, t)$ as follows:

$$\tilde{x}_0 = \frac{1}{\sqrt{\overline{\alpha}_t}}x_t - \frac{\sqrt{1-\overline{\alpha}_t}}{\sqrt{\overline{\alpha}_t}}\epsilon_\theta(x_t, t). \tag{3}$$

Cosine similarity is employed to measure the relative distances between the predicted images $\tilde{x}_0$. The probability distributions for $\tilde{x}_0^i$ ($0 \leq i \leq N$) in source and adapted models are as follows:

$$p_{sou}^{unc,i} = sfm(\left\{sim(\tilde{x}_{0_{sou}}^i, \tilde{x}_{0_{sou}}^j)\right\}_{\forall i \neq j}), \quad p_{ada}^{unc,i} = sfm(\left\{sim(\tilde{x}_{0_{ada}}^i, \tilde{x}_{0_{ada}}^j)\right\}_{\forall i \neq j}), \tag{4}$$

where $sim$ and $sfm$ denote cosine similarity and softmax function, respectively. However, it's difficult to build correspondence for T2I generation with fixed noise inputs since the source and

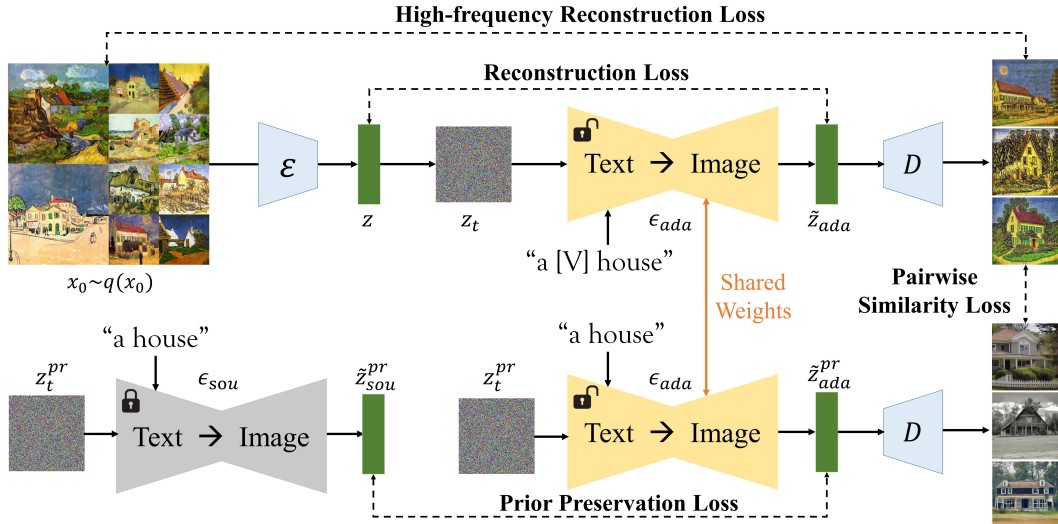

Figure 3: **Overview of DomainStudio on T2I DDPMs.** DomainStudio on unconditional DDPMs can be directly generalized to T2I DDPMs for domain-driven generation since it only uses image-level information for additional optimization. Prior preservation loss is used to preserve source samples during domain adaptation.

adapted samples have different conditions. We find that keeping similar distributions of relative pairwise distances for randomly generated source and adapted samples also maintains the distributions of subjects well for T2I generation. Given batches of noised source latent codes $\{z_t^{pr,n}\}_{n=0}^N$ and target latent codes $\{z_t^n\}_{n=0}^N$, we build probability distributions for source and adapted samples as follows:

$$p_{sou}^{text,i} = sfm(\Big\{sim(D(\tilde{z}_{ada}^{pr,i}), D(\tilde{z}_{ada}^{pr,j}))\Big\}_{\forall i \neq j}), \quad p_{ada}^{text,i} = sfm(\Big\{sim(D(\tilde{z}_{ada}^i), D(\tilde{z}_{ada}^j))\Big\}_{\forall i \neq j}), \quad (5)$$

where $\tilde{z}_{ada}^{pr}$ and $\tilde{z}_{ada}$ are denoised source and target latent codes produced by adapted models. The image-level pairwise similarity loss for unconditional and T2I generation are as follows:

$$\mathcal{L}_{img}^{unc} = \mathbb{E}_{t,x_0,\epsilon} \sum_i D_{KL}(p_{ada}^{unc,i} \| p_{sou}^{unc,i}), \tag{6}$$

$$\mathcal{L}_{img}^{text} = \mathbb{E}_{t,z_t,z_t^{pr},\epsilon,\epsilon^{pr}} \sum_i D_{KL}(p_{ada}^{text,i} \| p_{sou}^{text,i}), \tag{7}$$

where $D_{KL}$ represents KL-divergence and $z_t^{pr}$ represents the source latent codes added with noises $\epsilon^{pr}$. $\mathcal{L}_{img}$ prevents adapted samples from being too similar to each other or replicating training data. It encourages adapted models to keep the distributions of subjects in adapted samples similar to source samples and learns domain knowledge from training samples for domain-driven generation.

We are inspired by CDC (Ojha et al., 2021), which builds pairwise similarity loss based on features in certain layers of the generator in StyleGAN2 (Karras et al., 2020b) to preserve cross-domain similarity. DDPMs are trained to predict noises with multiple steps in synthesizing images. It is difficult to find proper features to represent generated samples. As a result, we directly build image-level pairwise similarity with the completely denoised samples, which makes our approach compatible with DDPMs. Besides, our method is designed for domain-driven generation to preserve the basic distributions of subjects, which take up most parts of the generated samples, instead of building one-to-one correspondence across domains like CDC. We use various noise inputs for source and adapted samples in T2I generation but still achieve the preservation of subject categories.

### 3.2 HIGH-FREQUENCY DETAILS ENHANCEMENT

To begin with, we employ the typical Haar wavelet transformation (Daubechies, 1990) to disentangle images into multiple frequency components. Haar wavelet transformation decomposes inputs into the low-frequency component $LL$ and high-frequency components $LH$, $HL$, and $HH$. We define $hf$ as the sum of all high-frequency components needing enhancement: $hf = LH + HL + HH$.

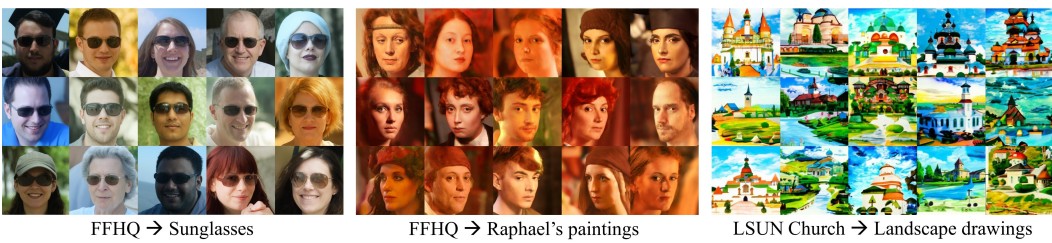

| FFHQ → Sunglasses | FFHQ → Raphael's paintings | LSUN Church → Landscape drawings |

Figure 4: DomainStudio generation samples under several 10-shot unconditional adaptation setups.

We implement high-frequency details enhancement from two perspectives. Firstly, we use the proposed pairwise similarity loss to preserve high-frequency details learned from source samples. The probability distributions for the high-frequency components of generated samples of source and adapted models for unconditional and T2I generation are as follows ($i \neq j$):

$$pf_{sou}^{unc,i} = sfm(\left\{ sim(hf(\tilde{x}_{0_{sou}}^i), hf(\tilde{x}_{0_{sou}}^j)) \right\}), pf_{ada}^{unc,i} = sfm(\left\{ sim(hf(\tilde{x}_{0_{ada}}^i), hf(\tilde{x}_{0_{ada}}^j)) \right\}), \quad (8)$$

$$pf_{sou}^{text,i} = sfm(\left\{ sim(hf(D(\tilde{z}_{ada}^{pr,i})), hf(D(\tilde{z}_{ada}^{pr,j}))) \right\}), \quad (9)$$

$$pf_{ada}^{text,i} = sfm(\left\{ sim(hf(D(\tilde{z}_{ada}^i)), hf(D(\tilde{z}_{ada}^j))) \right\}). \quad (10)$$

Similar to the image-level losses shown in Eq. 6 and 7, the pairwise similarity loss for the high-frequency components in unconditional and T2I generation are defined as follows:

$$\mathcal{L}_{hf}^{unc} = \mathbb{E}_{t,x_0,\epsilon} \sum_i D_{KL}(pf_{ada}^{unc,i} \,||\, pf_{sou}^{unc,i}), \quad (11)$$

$$\mathcal{L}_{hf}^{text} = \mathbb{E}_{t,z_t,z_t^{pr},\epsilon,\epsilon^{pr}} \sum_i D_{KL}(pf_{ada}^{text,i}||pf_{sou}^{text,i}). \quad (12)$$

Secondly, we propose high-frequency reconstruction loss to guide adapted models to learn more high-frequency details from limited training data by minimizing the mean squared error (MSE) between the high-frequency components in adapted training samples, which can be expressed for unconditional and T2I generation as follows:

$$\mathcal{L}_{hfmse}^{unc} = \mathbb{E}_{t,x_0,\epsilon} \left[ ||hf(\tilde{x}_{0_{ada}}) - hf(x_0)|| \right]^2, \quad (13)$$

$$\mathcal{L}_{hfmse}^{text} = \mathbb{E}_{t,x_0,z_t,\epsilon} ||hf(D(\tilde{z}_{ada})) - hf(x_0)||^2. \quad (14)$$

### 3.3 OVERALL OPTIMIZATION TARGET

The overall optimization target of DomainStudio combines all the methods proposed above to realize the preservation of subject distributions and high-frequency details enhancement. The loss function for unconditional and T2I generation are expressed in Eq. 15 and 16 respectively:

$$\mathcal{L}^{unc} = \mathcal{L}_{simple}^{unc} + \lambda_{11}\mathcal{L}_{vlb} + \lambda_2\mathcal{L}_{img}^{unc} + \lambda_3\mathcal{L}_{hf}^{unc} + \lambda_4\mathcal{L}_{hfmse}^{unc}, \quad (15)$$

$$\mathcal{L}^{text} = \mathcal{L}_{simple}^{text} + \lambda_{12}\mathcal{L}_{pr} + \lambda_2\mathcal{L}_{img}^{text} + \lambda_3\mathcal{L}_{hf}^{text} + \lambda_4\mathcal{L}_{hfmse}^{text}. \quad (16)$$

We set the weight $\lambda_{11}$ of the variational lower bound loss $\mathcal{L}_{vlb}$ (Nichol & Dhariwal, 2021) as 0.001 and the weight $\lambda_{12}$ of the prior preservation loss $\mathcal{L}_{pr}$ (Ruiz et al., 2023) as 1 (see more details in Appendix B). We empirically find $\lambda_2, \lambda_3$ ranging between 0.1 and 1.0 and $\lambda_4$ ranging between 0.01 and 0.08 to be effective for most unconditional adaptation setups and $\lambda_2, \lambda_3$ ranging between 100 and 500 and $\lambda_4$ ranging between 0.1 and 1.0 to be effective for most T2I adaptation setups.

## 4 EXPERIMENTS

We evaluate DomainStudio with a series of domain-driven generation tasks using extremely limited data ($\leq 10$ images). It is compared with directly fine-tuned DDPMs, GAN-based few-shot image generation methods, and few-shot fine-tuning methods of large-scale T2I models on generation quality and diversity qualitatively and quantitatively. Ablation analysis is discussed in Sec. 4.3.

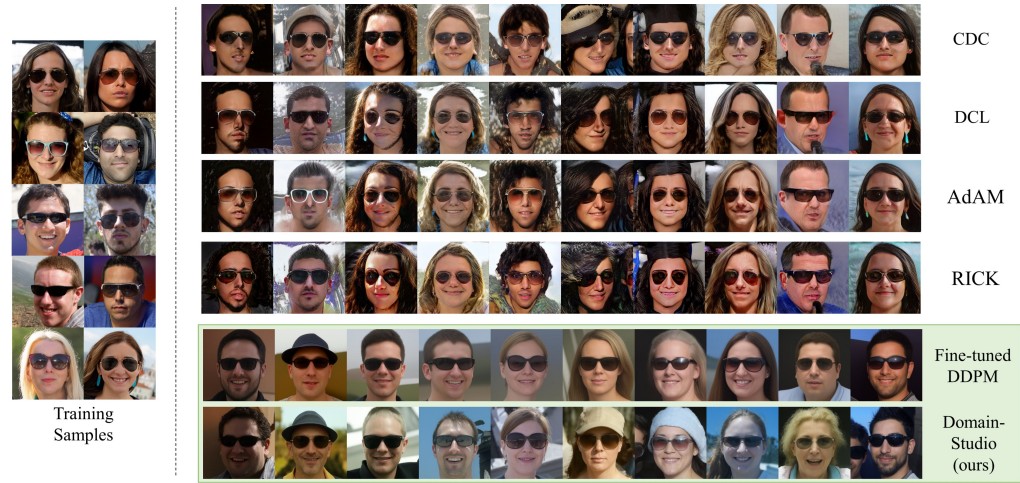

Figure 5: 10-shot unconditional image generation samples on FFHQ → Sunglasses. All the samples of GAN-based approaches are synthesized from fixed noise inputs (rows 1-4). Samples of the directly fine-tuned DDPM and DomainStudio are synthesized from fixed noise inputs as well (rows 5-6). Our approach generates high-quality results with fewer blurs and artifacts and achieves considerable generation diversity.

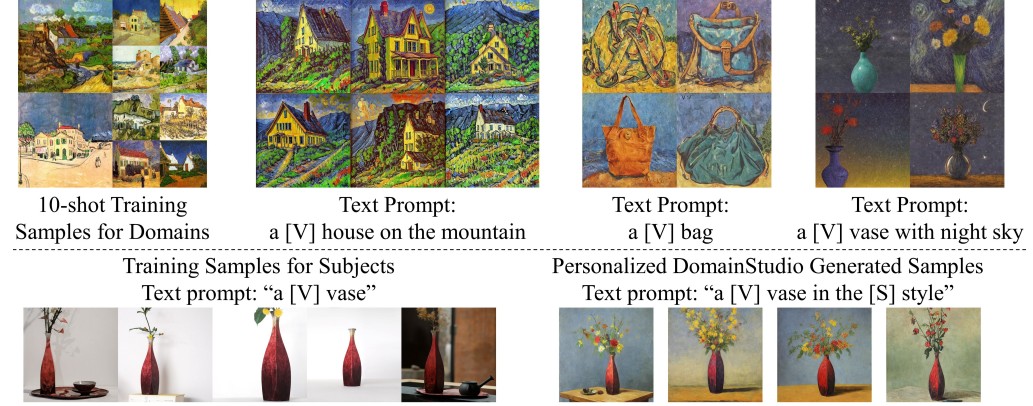

Figure 6: Given 10-shot training samples to characterize the target domain, DomainStudio can synthesize samples in target domains with diverse contexts, different categories of subjects or personalized subjects.

**Basic Setups** We train DDPMs from scratch on FFHQ (Karras et al., 2020b) and LSUN Church (Yu et al., 2015) as source models for unconditional generation. Stable Diffusion (Rombach et al., 2022) v1.4 is employed as the source model for T2I generation. Several datasets containing less than 10 samples are used as training datasets. More details of implementation are added in Appendix H.

**Baselines** Since few prior works realize DDPM-based few-shot image generation unconditionally, we employ several GAN-based baselines sharing similar targets with us on subject-consistent source/training datasets: CDC (Ojha et al., 2021), DCL (Zhao et al., 2022b), AdAM (Zhao et al., 2022a), and RICK (Zhao et al., 2023). All the GAN-based methods are implemented based on StyleGAN2 (Karras et al., 2020b). The StyleGAN2 models and unconditional DDPMs trained on the large source datasets share similar generation quality and diversity (see more details in Appendix I). Modern few-shot fine-tuning methods Textual Inversion (Gal et al., 2022) and DreamBooth (Ruiz et al., 2023) are used as baselines for T2I generation. Textual Inversion is trained to learn styles.

**Evaluation Metrics** We follow CDC to employ Intra-LPIPS for diversity evaluation and FID (Heusel et al., 2017) for quality evaluation. As for T2I generation, we employ CLIP (Radford et al., 2021) to measure the textual alignment with text prompts using CLIP-Text, which is the average pairwise cosine similarity between the CLIP embeddings of text prompts and generated samples and evaluates the preservation of subjects. More details of metrics are added in Appendix C.

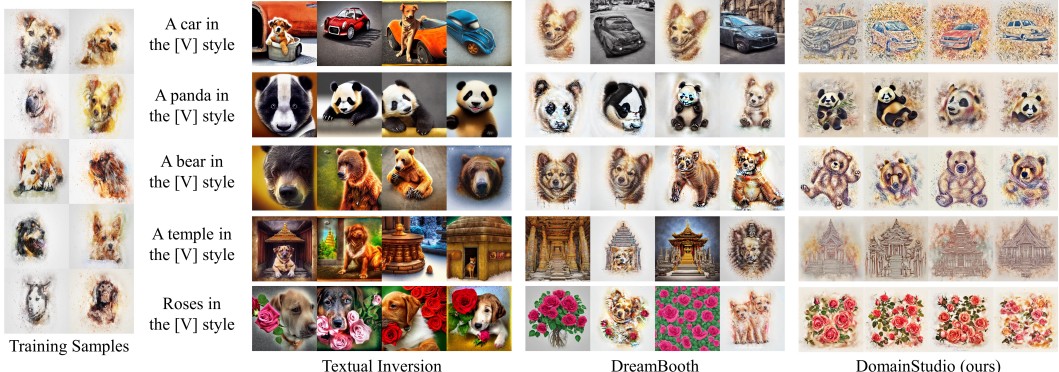

Figure 7: 10-shot T2I generation samples trained on Watercolor dogs. DomainStudio produces samples containing diverse subjects consistent with text prompts and sharing the same style with training data.

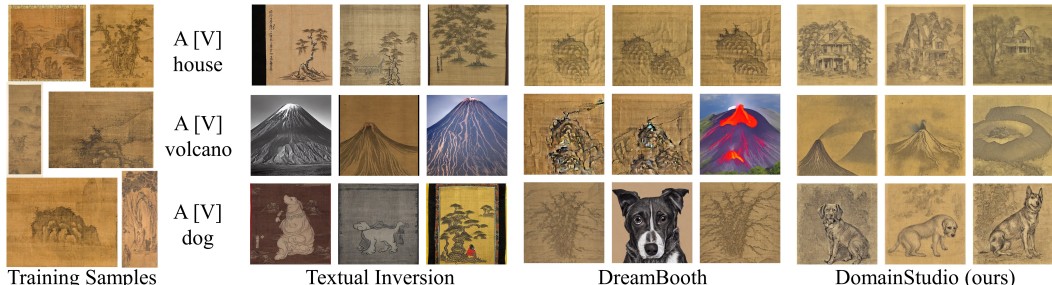

Figure 8: T2I generation samples trained on 6-shot Ink paintings. DomainStudio adapts subjects mentioned in text prompts to target domains naturally while baselines get unrealistic results due to underfitting or overfitting.

## 4.1 QUALITATIVE EVALUATION

**Unconditional Image Generation** As shown in Fig. 4, DomainStudio adapts source models to target domains naturally and produces diverse samples under several 10-shot adaptation setups. Samples of diverse faces can be found when adapting FFHQ to target domains. The adaptation from LSUN Church to Landscape drawings retains various church structures. Fig. 5 shows samples of DomainStudio and baselines on 10-shot FFHQ → Sunglasses. GAN-based baselines generate some incomplete sunglasses and unnatural blurs and artifacts. Directly fine-tuned DDPM produces smoother results but lacks details like lighting effects and fine hair structures. In contrast, DomainStudio improves generation quality and diversity and achieves more pleasing visual effects.

**T2I Generation** As illustrated in Fig. 1 and 6, DomainStudio is qualified for domain-driven generation regardless of whether the category of subjects in training samples is consistent with adapted samples. For example, given 10-shot Van Gogh houses as training data, we synthesize samples with text prompts such as "a [V] bag". Besides, we combine DomainStudio with DreamBooth to realize domain-driven generation with personalized subjects (see more details in Appendix G).

| Metrics | FID (↓) | | Intra-LPIPS (↑) | | |
|---------|---------|---|---|---|---|
| Datasets | Babies | Sunglasses | Babies | Sunglasses | Sketches |
| CDC | 74.39 | 42.13 | $0.583 \pm 0.014$ | $0.579 \pm 0.018$ | $0.454 \pm 0.017$ |
| DCL | 52.56 | 38.01 | $0.579 \pm 0.018$ | $0.574 \pm 0.007$ | $0.461 \pm 0.021$ |
| AdAM | 48.43 | 28.03 | $0.573 \pm 0.016$ | $0.559 \pm 0.017$ | $0.424 \pm 0.018$ |
| RICK | **39.39** | **25.22** | $0.589 \pm 0.010$ | $0.591 \pm 0.030$ | $0.443 \pm 0.025$ |
| Fine-tuned DDPMs | 114.95 | 54.47 | $0.513 \pm 0.026$ | $0.527 \pm 0.024$ | $0.473 \pm 0.022$ |
| DomainStudio (ours) | 48.92 | 34.75 | $\mathbf{0.599 \pm 0.024}$ | $\mathbf{0.604 \pm 0.014}$ | $\mathbf{0.495 \pm 0.024}$ |

Table 1: Quantitative results of unconditional few-shot image generation (source datasets: FFHQ). DomainStudio achieves better generation diversity than directly fine-tuned DDPMs and prior GAN-based methods.

DomainStudio is compared with Textual Inversion and DreamBooth in Fig. 7 and 8. It's difficult for baselines to learn the domain knowledge from few-shot data, especially when the subjects in adapted samples are different from training samples. For instance, when generating temples and roses in the watercolor style, baselines tend to combine the subjects in training samples with the subjects mentioned in text prompts. They lack guidance on what to learn from few-shot data and what to preserve from source models, resulting in overfitting or underfitting results. DomainStudio successfully adapts the subjects mentioned in text prompts to the domain of training samples, achieving high-quality and diverse samples. More visualized results are provided in Appendix M.

## 4.2 QUANTITATIVE EVALUATION

We provide the quantitative results of DomainStudio on unconditional and T2I generation in Table 1 and 2. DomainStudio achieves a superior improvement of generation diversity compared with directly fine-tuned DDPMs and outperforms state-of-the-art GAN-based approaches in terms of Intra-LPIPS. Although DomainStudio fails to achieve the best FID, it produces samples in target domains with fewer blurs

| Methods | Van Gogh houses | Ink painting volcanoes |
|---|---|---|
| Textual Inversion | $0.259 \pm 0.001$ | $0.244 \pm 0.001$ |
| DreamBooth | $0.262 \pm 0.002$ | $0.275 \pm 0.004$ |
| DomainStudio (ours) | $\mathbf{0.276 \pm 0.002}$ | $\mathbf{0.301 \pm 0.002}$ |

Table 2: CLIP-Text (↑) results of DomainStudio on T2I tasks. DomainStudio outperforms baselines on text alignment.

and artifacts, resulting in better visual effects, as shown in the qualitative results. For T2I generation, DomainStudio achieves state-of-the-art CLIP-Text results, indicating its ability to preserve subjects in domain-driven generation and synthesize images consistent with text prompts while adapting to target domains. More quantitative results and analysis are added in Appendix D.

## 4.3 ABLATION ANALYSIS

We ablate our approach to show the roles of two parts. Subject distributions preservation is the basis of DomainStudio, without which mode collapse occurs and leads to blurred and low-quality results. High-frequency details enhancement guides adapted models to generate more details (e.g., house structures and hairstyles). DomainStudio combines them to achieve compelling and diverse results and better metrics. Without these two parts, DomainStudio degrades to DreamBooth and directly fine-tuned models in T2I and unconditional generation. More detailed ablations are provided in Appendix E.

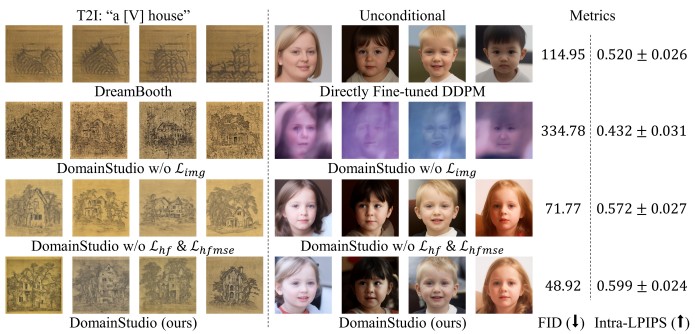

Figure 9: Ablation analysis of DomainStudio on T2I (houses in the ink painting style) and unconditional DDPMs (FFHQ → Babies).

## 5 CONCLUSION

We propose DomainStudio, a novel approach to realize few-shot and domain-driven image generation. DomainStudio fills the vacuum of synthesizing samples in specific domains (e.g., styles) with DDPMs. It is compatible with both unconditional and T2I DDPMs. DomainStudio first introduces DDPMs to unconditional image generation given limited data. It produces compelling results with rich details and few blurs, outperforming current state-of-the-art GAN-based methods on visual effects and generation diversity. DomainStudio also performs better than prior few-shot fine-tuning methods in T2I domain-driven generation. It is qualified for domain-driven T2I tasks regardless of the subject gaps between source and training samples. This work takes an essential step toward more data-efficient DDPMs. The limitations and societal impacts are discussed in Appendix F.

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
