# OpenReview forum: "DomainStudio: Fine-Tuning Diffusion Models for Domain-Driven Image Generation using Limited Data"
_ICLR.cc/2024/Conference — ICLR 2024 Conference Withdrawn Submission_

### Official Review · Reviewer_w5yc · 2023-10-28

**Soundness:** 4 excellent
**Presentation:** 4 excellent
**Contribution:** 3 good
**Rating:** 6
**Confidence:** 4

**Summary:**

This paper introduces a new domain-driven generation approach called DomainStudio, which aims to learn the features of target domains while maintaining diversity using limited data.
The key components of this approach are relative distance preservation and high-frequency detail enhancement. To achieve better diversity, the relative distance preservation is proposed, which uses pairwise similarity loss to keep the relative pairwise distances between generated samples similar to source samples. Additionally, high-frequency detail enhancement is proposed to address the lack of high-frequency details in generated results. This enhancement guides the adapted model to learn more high-frequency details from limited training data and source domains.
The paper investigates both unconditional image generation and text-to-image generation methods and conducts comprehensive experiments to demonstrate the effectiveness of DomainStudio. DomainStudio outperforms current state-of-the-art unconditional GAN-based approaches and conditional DDPM-based text-to-image approaches in terms of both generation quality and diversity.

**Strengths:**

1.	This paper is well-written, with a clear motivation and definition of domain-driven generation.
2.	Comprehensive experiments were conducted to evaluate the performance of both unconditional and conditional DDPMs fine-tuned on limited data, revealing a problem with existing state-of-the-art generative models. The experiments in Section 5 effectively demonstrate the effectiveness of the proposed method and each key component of the paper.
3.	The performance of this paper is significant, as it achieves better generation quality and diversity than current state-of-the-art unconditional GAN-based approaches and conditional DDPM-based text-to-image approaches.

**Weaknesses:**

The relative distance preservation is not the first time to be proposed. The cross-domain correspondence proposed in the paper “Few-shot image generation via cross-domain correspondence” is very similar to the one in this paper.

**Questions:**

1.	As mentioned earlier, relative distance preservation has been utilized in GAN-based methods. Could you please explain the differences between the relative distance preservation approach in this paper and the cross-domain correspondence approach in "Few-shot image generation via cross-domain correspondence"?
2.	Assuming that StyleGAN2 models and DDPMs trained on large source datasets share similar generation quality and diversity, why does this paper outperform the CDC paper in terms of performance?
3.	In Section 4, what is the reason behind the statement that "Subjects in source and target domains do not need to be consistent with training samples"?
4.	Could you please explain why high-frequency enhancement not only improves image details but also enhances diversity, as shown in Tables 6 and 7? It appears that high-frequency enhancement and relative distance preservation have similar loss functions, resulting in similar effects.
5.	Are there any DDPM models for few-shot image generation other than subject-driven series methods? If so, what is their performance?

---

> ### Author Response · Authors · 2023-11-15
> **Response from authors**
>
> Thanks for your precious time for reviews. Here we provide corresponding responses and results to cover your concerns.
>
> 1.$\textbf{Cross-domain correspondence approach}$: We provided the discussion between our approach and this prior work in Appendix J. In the revised manuscript, we illustrate the difference in Sec 3.1 as follows:
>
>     We are inspired by CDC, which builds pairwise similarity loss based on features in certain layers of the generator in StyleGAN2 to preserve cross-domain similarity. DDPMs are trained to predict noises with multiple steps in synthesizing images. It is difficult to find proper features to represent generated samples. As a result, we directly build image-level pairwise similarity with the completely denoised samples, which makes our approach compatible with DDPMs. Besides, our method is designed for domain-driven generation to preserve the basic distributions of subjects, which take up most parts of the generated samples, instead of building one-to-one correspondence across domains like CDC. We use various noise inputs for source and adapted samples in T2I generation but still achieve the preservation of subject categories.
>
> More detailed discussion of our approach is added in Appendix J.
>
> 2.$\textbf{Outperforming CDC}$: As discussed above, our approach is different from CDC. Besides, CDC is based on StyleGAN2 while our approach is designed for DDPMs. These models are different. We cannot simply get an equation like “StyleGAN2 + CDC == DDPM + CDC“ even if the source models share similar metrics of FID and Intra-LPIPS. As shown in the visualized results, unexpected blurs and deformation are inevitable in the adapted samples of GAN-based approaches. However, such problems do not exist for DDPMs. Directly fine-tuned DDPMs get smooth results with limited diversity and lack high-frequency details. With our approach, adapted DDPMs achieve high-quality results with rich details.
>
> 3.$\textbf{Subjects in source and target domains}$: Our approach is designed for domain-driven generation by preserving the subjects in source samples and learning the domain knowledge from training samples. We have revised this sentence as: “Subjects in source and adapted samples share the same category but can be different from subjects in training samples.” to make it clearer. For example, given 10-shot Van Gogh houses paintings, we can set the source samples with the prompt “a dog” and the adapted samples with the prompt “a [V] dog”. The subjects in source and adapted samples are different from the subject “a house” in training samples. We have revised the whole paper to remove the confusing description of “source domains” and “target samples” to avoid similar misunderstanding.
>
> Several common expressions used in our paper:
>
> “source samples”: samples produced by the pre-trained model, we aim to preserve the subjects in them
>
> “target domains”: domains (e.g., styles) extracted from few-shot training data
>
> “adapted samples”: samples produced by adapted models, which are designed to preserve the same category of subjects in source samples and be adapted to target domains
>
> 4.$\textbf{Details and Diversity}$: The high-frequency enhancement in our approach preserves more diverse details in adapted samples. It is designed from two perspectives: preserving more details provided by source models (which share the same format of pairwise similarity loss) and learning more details from limited data for finer quality. More details preservation contributes to generation diversity as well. For example, we have more diverse detailed structures of houses in the samples of full DomainStudio compared with DomainStudio w/o high-frequency enhancement, which tends to synthesize simpler houses (see Fig. 9 in the revised manuscript).
>
> 5.$\textbf{Other Methods}$: As far as we know, we are the first to tackle synthesizing samples in specific domains using limited data with DDPMs.
>
> Please let us know if our responses solve your concerns. If you still have any unclear parts about our work, please let us know as well. Thanks for your review.

---

> ### Author Response · Authors · 2023-11-22
>
> Dear reviewer w5yc,
>
> We thank you for your valuable reviews. We have revised the manuscript and provided detailed response to cover your concerns. Today is the last day for discussion between authors and reviewers. We are looking forward to your reply.
>
> Thanks for your review!

---

### Official Review · Reviewer_4uc5 · 2023-10-28

**Soundness:** 3 good
**Presentation:** 3 good
**Contribution:** 3 good
**Rating:** 8
**Confidence:** 4

**Summary:**

The paper introduces DomainStudio, a strategy for fine-tuning pretrained diffusion models towards very small adaptation datasets (no more than 10 images), without losing the ability to generate diverse and highly detailed images. The proposed strategy can work on either unconditional image generation aimed at matching the style of the fine-tuning images, or text-conditioned image generation to generate images that match the style of the fine-tuning images while also matching specific structure or content described in the text prompt.

The main idea of the method is the introduction of two regularization losses to supplement the standard diffusion training loss. One of these losses is designed to promote diversity in the predicted images, and the other is designed to promote high-frequency details. Combined, these regularizers produce noticeable improvement even compared to DreamBooth and other high-quality baselines, particularly for settings where there are very few images provided for fine-tuning.

**Strengths:**

Overall the paper is well-written and easy to follow, and the results are visually compelling.

**Weaknesses:**

Substantial suggestions for improvement
- Probably my most significant concern/suggestion is to include the quantitative evaluation results in the main paper, rather than putting them in the supplement. This is a core way that future papers will be able to build on your work and compare to it. I realize that in this field the metrics aren’t always super well-aligned with human perception of quality, so they may not be too meaningful, but in my opinion it would be better to include them and explain any caveats to the metrics rather than sweeping all of it under the rug (to the appendix).
- Likewise, I would strongly encourage the authors to include the ablation study results in the main paper. This is the only real justification for introducing two new losses; it merits inclusion in the main text.
- The introduction is long and the first half of it reads like a related work section. I would prefer if the authors more directly introduce the main idea of their contribution, and this would also save space that could be put towards the quantitative results and ablations. When citing related work, especially in the introduction, I would also encourage the authors to mention the name of the paper (e.g. DreamBooth) so that readers don’t need to flip back and forth to the references to understand what is being described.
- The related work section could also be improved and shortened. Currently it reads like a survey with each related paper described by a sentence, but not much higher-level analysis to help the reader group the related works into relevant categories and understand how these categories differ from each other and from the proposed method.

**Questions:**

Minor suggestions for improvement
- Figure 1 caption describes samples containing the “same subject as training samples”, but the examples shown (“a [V] house”) do not look like they are actually preserving the “identity” of a specific house from the training set.
- If you want a citation to not interrupt the flow of a sentence, you can use \citep rather than \citet. If you want to use the citation as a noun in a sentence, use \citet.
- The “Text-to-image Generation” paragraph describes DreamBooth as a “light-weight fine-tuning method”, but my impression was that DreamBooth is relatively expensive as a fine-tuning method because it edits the full weights.
- I don’t really buy the motivation for the high-frequency regularization—the sample images in Figure 2 and Figure 3 don’t strike me as particularly low-detail. Including the ablation in the main text would help motivate this loss, or perhaps there are other examples you can show to motivate why this loss is needed.
- It would be nice if the captions (or figures) for Figure 5 and Figure 6 more directly described the novelty of the contribution, and how the two figures fit together, since these are the main visual illustration of the method.

---

> ### Author Response · Authors · 2023-11-15
> **Response from authors**
>
> Thanks for the so many valuable suggestions provided in your review. We have revised our paper following your valuable suggestions.
>
> 1.$\textbf{Quantitative Results}$: We have moved the key part of the quantitative evaluation to the main paper. More details are provided in the appendix.
>
> 2.$\textbf{Ablation Analysis}$: We have moved the representative parts of the ablation analysis to the main paper. More details are provided in the appendix.
>
> 3.$\textbf{Introduction}$: We have revised the introduction part to give a clear definition of “domain-driven generation” and the necessity of the few-shot domain-driven task. Then, the main idea of contribution is introduced. We directly use the name for related works like DreamBooth and Textual Inversion.
>
> 4.$\textbf{Related Work}$: We have revised related work to provide much more higher-level analysis to identify the differences between our approach and prior works.
>
> 5.$\textbf{Fig.1}$: The same subject refers to the same category of subjects. We have revised Fig.1 and the whole paper to provide a clearer definition of our task to avoid misunderstanding.
>
> 6.$\textbf{Citation}$: We have revised all our paper's citations and changed them to \citep. Thanks!
>
> 7.$\textbf{Lightweight}$: It is a writing mistake, and we have fixed this issue in the revised manuscript.
>
> 8.$\textbf{High-frequency Regularization}$: We find Sec. 3 is not convincing and removed this part in the revised manuscript. We include the ablation analysis to demonstrate the motivation of each part in our approach.
>
> 9.$\textbf{Fig.5 and Fig.6}$: We have revised Fig.5 to make it fit with Fig.6 better and revised their captions to describe the contributions.
>
> Looking forward to more discussion and suggestions on our paper from you. Thanks again!

---

> > ### Comment · Reviewer_4uc5 · 2023-11-22
> >
> > Thanks for making these changes in the revision; I think the paper is definitely improved (and have increased my score)! Most of my concerns are addressed in the revision, and I also appreciated the clarification of the task setting and its motivation even though that was requested by a different reviewer. One remaining suggestion is that I would prefer if the ablation included separate results with each of the two regularizers rather than only showing both or neither, since the point of an ablation is to understand the impact of different components of your approach individually.

---

> > > ### Author Response · Authors · 2023-11-22
> > >
> > > Thanks for your positive feedback and valuable suggestions. We have revised the ablation part in the revised manuscript following your suggestion. Our approach is composed of subject distributions preservation and high-frequency details enhancement. We add ablation analysis to show that subject distributions preservation is the basis of our approach, without that mode collapse occurs and leads to very blurred and low-quality results. Applying high-frequency details enhancement only cannot produce reasonable results.
> > >
> > > Thank you again for all the valuable suggestions and your precious time in reviewing this paper sincerely!

---

### Official Review · Reviewer_mvse · 2023-11-01

**Soundness:** 2 fair
**Presentation:** 2 fair
**Contribution:** 1 poor
**Rating:** 3
**Confidence:** 5

**Summary:**

This work proposes an approach for adapting the pre-trained denoising diffusion probabilistic models (DDPMs) to a target domain using limited data. It discussed three major contributions:
- analyzing the shortcomings of adapting DDPMs to target domains with limited data
- proposing a pairwise loss to keep the diversity of the image generation in the adapted generative model
- exploiting the frequency components for improving the generation quality
This work also claims to be the first one that achieves better generation performance for few-shot image generation (e.g., 10-shot) compared to SOTA works based on the GAN.

**Strengths:**

This work aims to address a crucial aspect of image generation with limited data in DDPMs. Even though this is studied in a length for GANs, there is a lack of efficient approaches for DDPMs under limited data or few-shot setups.

**Weaknesses:**

I have major concerns regarding the current version of this work:

I. All of the three contributions mentioned in the paper, have different types of flaws and problems:

a) Section 3 aims to study the adaptation of a pre-trained DDPM to a target domain using limited data for two different types of
generations: unconditional generation and text-to-image generation. However, this part has nothing new, and lacks enough details:

- The major problem with this section is that this phenomenon is quite well-known in generative models, and simply running this for DDPMs is not adding any value in my opinion, and this can not be considered as a contribution!

- In Figure 2, the details of measuring the similarity are missing. For example, which features are used to compute the cosine similarity between two images?
In addition, this study is not enough. Using only two samples to discuss the diversity and the quality is not a technical way to discuss this and a more systematic approach for this is required (e.g., using quantitative measurements like FID for generated data).

- In Figure 3, the details are also missing. In addition, the approach used for the comparison (DreamBooth) is not related to the proposed method in terms of the task as this approach aims to implant a subject in a pre-trained model by linking it to a unique identifier.
The question is how Dreambooth is going to assign all these varied inputs into an identifier [v]? In this Figure, the Dreambooth is asked to learn the target domain distribution however it was originally designed to implant subjects in the same domain.

- Figure 3 gives a very bad impression that it is not clear to the authors which task they do have in mind, and they mix some unrelated concepts (generative tasks; subject diven generation vs domain adaptation; see [1]) together!

b) The pairwise loss proposed in sec. 4.1. is identical to the main idea proposed in the cross-domain correspondence (CDC) [CVPR'21] proposed for GANs, and this work fails to discuss the related work, borrowing the idea from CDC, and the possible new parts added for DDPMs.

c) Similarly, the third contribution, has been studied before in the literature in FreGAN [2] and MaskedGAN [3] and this work fails to discuss the related work and what is different in this paper.

II. My second concern is regarding the clarity of the tasks that this work aims to address. The unconditional generation task is clear from similar studies in the literature. But for the second task:
- the definition of the text-to-image generative domain adaptation is not clear to me, and there is no clear and technical definition in this paper
- This task seems to be **unnecessary**, as the pre-trained text-to-image model can handle this task and there is no need for adaptation. For the running example in most parts of the paper (e.g., the house painting by Van Gogh), the pre-trained text-to-image model (latent/stable diffusion) generates the required images using the proper prompt. For example, one can pass "The Yellow House in Vincent Van Gogh painting style" to stable diffusion / DALL.E and get much better results (I personally did) than what is shown in Figures 1 and 2.


III. The final concern is regarding the experimental results:

- For the first task (unconditional generation), the quantitative evaluation is shifted to the supp. which in my opinion is the most critical part that shows your approach can outperform state-of-the-art (as claimed in the abstract).
Checking the results in supp. (Tables 2 and 3 of supp), this paper only compared with old approached (DCL from CVPR'22) and excludes the recent works (AdAM from NeurIPS'22, and RICK from CVPR'23) that have much better performance than the proposed method. Including these two methods can clarify how efficient the proposed method is.

- The visual results provided in Figure 6 (for Raphael's painting) is not very appealing. The generator can not adapt to the target domain in this case and is almost similar to the source domain FFHQ.

- For the second task, as the pre-trained text-to-image model can already generate some images using a proper text prompt (as discussed before) experimental results should justify the necessity of tasks in terms of either failing the pre-trained model, or better performance using the proposed method.

I am willing to increase my scores if you can provide enough details regarding these weaknesses.

**References:**

- [1] A survey on generative modeling under limited data, few shots, and zero shot

- [2] FreGAN: Exploiting Frequency Components for Training GANs under Limited Data

- [3] Masked generative adversarial networks are data-efficient generation learners

**Questions:**

Please refer to weaknesses.

---

> ### Author Response · Authors · 2023-11-15
> **Response from authors (Contributions and Sec.3)**
>
> Thanks for your precious time for reviews. We provide corresponding responses here and revise our paper to cover your concerns.
>
> 1.$\textbf{Section3}$: We agree with you that Section 3 is not surprising and placing this part in the main paper makes our task confusing. We have provided a clear definition of the domain-driven generation task in the abstract and introduction of the revised manuscript and removed this part (Sec.3) in the revised manuscript.
>
> 2.$\textbf{Fig.2}$: In the original Fig.2, we directly use image-level information (RGB values) to compute cosine similarity. Fig.2 has been removed in the revised manuscript. Instead, we use ablations of our approach with qualitative and quantitative (FID and Intra-LPIPS) results (which are originally provided in Appendix D) to discuss the diversity and high-frequency details that are improved by our approach compared with directly fine-tuned models.
>
> 3.$\textbf{Fig.3}$: In the original Fig. 3, we use DreamBooth as a comparison to demonstrate that it is not qualified for domain-driven tasks or it cannot learn the domain knowledge with an identifier [V], which is achieved by our approach. DreamBooth w/o prior preservation loss corresponds to directly fine-tuned Stable Diffusion. Without the proposed relative distances preservation and high-frequency details enhancement, DomainStudio degrades to DreamBooth on text-to-image generation. Overall, we agree with you that Fig.3 makes readers confusing and removed it in the revised manuscript.
>
> 4.$\textbf{Related Works}$: We put the discussion between our approach and CDC in Appendix J. In the revised manuscript, we illustrate the difference in Sec 3.1 as follows:
>
>     We are inspired by CDC, which builds pairwise similarity loss based on features in certain layers of the generator in StyleGAN2 to preserve cross-domain similarity. DDPMs are trained to predict noises with multiple steps in synthesizing images. It is difficult to find proper features to represent generated samples. As a result, we directly build image-level pairwise similarity with the completely denoised samples, which makes our approach compatible with DDPMs. Besides, our method is designed for domain-driven generation to preserve the basic distributions of subjects, which take up most parts of the generated samples, instead of building one-to-one correspondence across domains like CDC. We use various noise inputs for source and adapted samples in T2I generation but still achieve the preservation of subject categories.
>
> We have added the discussion between high-frequency details enhancement in our approach with related works, including MaskedGAN and FreGAN, as an added part of related work (Sec. 2) in the revised manuscript. Our approach differs from the prior GAN-based methods in many aspects.
>
> 5.$\textbf{Contributions}$: We have revised the contributions of our work in the revised manuscript. The contributions of our work come from the task definition, the domain-driven generation approach compatible with both unconditional and T2I models, and the effectiveness demonstrated with a series of experiments.

---

> ### Author Response · Authors · 2023-11-15
> **Response from authors (Task)**
>
> 6.$\textbf{Task Definition}$: This paper tackles a domain-driven generation task, which is defined as retraining the category of subjects in source samples and adapting them to target domains extracted from few-shot training data. It is designed to synthesize samples in specific domains (e.g., styles). We have provided detailed definition in the revised manuscript and show several application scenarios, some of which cannot be handled by existing approaches in Fig. 1 in the revised manuscript. When the source samples share the same class of subjects with training data, it shares the same target as prior unconditional few-shot image generation tasks (e.g., FFHQ $\rightarrow$ Sunglasses). When different, DomainStudio is designed to preserve subjects in source samples and learn domain knowledge only from training samples.
>
> 7.$\textbf{Task Necessity}$: Our work tackles the task of synthesizing samples in specific domains given few-shot training samples. It is necessary due to two reasons. Firstly, it is difficult to describe domains with text prompts accurately for T2I models due to the ambiguities in natural language. Besides, even if pre-trained models have captured typical domains owing to the presence of corresponding samples in their training data, they may still produce biased results influenced by the out-of-distribution effects. For example, there exist different styles in Van Gogh's paintings. Given a text prompt of ``Van Gogh style", T2I models may choose a certain style randomly or mix some of them, leading to unsatisfactory outputs. Therefore, DomainStudio is designed to synthesize samples in specific domains learned from the given few-shot training samples. We have added this part to the introduction of the revised manuscript to make our task setting more convincing.

---

> ### Author Response · Authors · 2023-11-15
> **Response from authors (Experiments)**
>
> 8.$\textbf{Additional Baselines}$: We have added the qualitative and quantitative results of AdAM and RICK, which are designed to deal with incompatible source/training samples, like FFHQ $\rightarrow$ AFHQ CAT. We run the official implementation code to get qualitative results and evaluate the FID and Intra-LPIPS results. We got similar visual effects as shown in their paper but failed to reproduce the FID results as well as reported in their paper. As for Intra-LPIPS (generation diversity), unconditional DomainStudio achieves better performance. We add FID results in their paper to the revised manuscript. Despite its outstanding FID results, RICK still fails to avoid generating unnatural deformation and blurs like prior GAN-based methods. Our approach shows apparently better visual effects, as shown in the visualized samples provided in the revised manuscript (Fig. 5 and Appendix N). We also fix the statement in abstract to “better visual effects than state-of-the-art GAN-based approaches”.
>
> 9.$\textbf{Raphael's paintings}$: We conduct additional experiments with weaker regularization and get better results in target domains on FFHQ $\rightarrow$ Raphael’s paintings and LSUN Church $\rightarrow$ Landscape drawings. Corresponding visualized and quantitative results have been updated in the revised manuscript.
>
> 10.$\textbf{Experiments necessity}$: As discussed above, taking the 10-shot landscape drawings (see scene 2 in Fig.1) as an example, it is hard to find proper text prompts for such data. There can be thousands (even more) of drawing styles in the training data of large-scale text-to-image models.
>
> Please let us know if our responses solve your concerns. If you still have any unclear parts about our work, please let us know as well. Thanks for your valuable suggestions.

---

> ### Author Response · Authors · 2023-11-22
>
> Dear reviewer mvse,
>
> We thank you for your valuable reviews. We have revised the manuscript and provided detailed response to cover your concerns. Today is the last day for discussion between authors and reviewers. We are looking forward to your reply.
>
> Thanks for your review!

---

> ### Comment · Reviewer_mvse · 2023-11-22
>
> I really appreciate the effort of the authors to address my concerns. However, I think there are still some major problems:
>
> I. Task Definition in Figure 1, is more intuitive and not accurate enough.
>
> - How can we claim that few-shot image generation (FSIG) learns subjects from the training samples given that there are only 10 subjects? note that these methods can generate thousands of subjects after adaptation.
>
> - The definition provided for the domain-driven image generation (preserve subject from prior knowledge, learn domain/style from training samples) is in fact the goal of most of the FSIG approaches that aim to generate images from close domains. For example, EWC [a] and CDC [b] motivate their approach with this goal.
>
> - Scene 4 in the figure is quite self-contradictory. First, the authors mention that the goal is to preserve the subject from the prior knowledge, but here they aim to personalize the subject from training samples. Am I missing something?
> In addition, the reported results do not preserve the subject at all. The subject in the training samples is a dog with the Chow Chow breed, and the generated ones are from other breeds.
>
> I think the authors need to think a bit more about the exact definition of the task that they have proposed, and the definition is still not clear and acceptable.
>
> II. I still am not convinced about the necessity of this task, as I mentioned in the main review. I was able to produce quite convincing results about the desired domain using just a proper text prompt (without any prompt engineering and just a hard text prompt). I suggest authors also think about this aspect more carefully and have a more systematic study/comparison with the hard prompt version.
>
> III. Regarding experimental results, I understand that it might be hard to replicate the results for SOTA approaches given the time limitations in rebuttal. However, the reported results in these papers, both quantitatively, and qualitatively (for example figure 25 in AdAM's Supp. for FFHQ $\rightarrow$ Sunglasses) seem to be much better than the provided results in this paper.
>
> **Overall**, I appreciate the effort from the authors and I believe the revised version has improved a lot. However, there are still some major problems that need to be thought about and addressed carefully. Hope my comments to be constructive for the next version of this work.

---

> ### Author Response · Authors · 2023-11-22
>
> Thanks for your detailed response. We still need to clarify several points about your concerns.
>
> 1.$\textbf{Task Definition}$:
> 1. FSIG tasks aim to produce diverse samples following the distributions of few-shot training samples. As illustrated in the revised related works, they share the same target with our work when the source and target domains share the same category of subjects, e.g, FFHQ --> Sunglasses. The subjects mentioned in our paper refer to the category of subjects, which has been illustrated in the revised manuscript. "Subjects learned from training samples" doesn't means that they are trained to preserve the 10 subjects in training samples. We have revised the table in Fig.1 in the manuscript to make it clearer.
>
> The target of FSIG is different from our work when subjects in source and training samples are different. For example, when FSIG adapts source models pre-trained on LSUN Church to 10-shot landscape drawings, they tend to produce samples without churches since there are no churches in training samples, which is the category of subjects in source samples (visualized results provided in Fig. 28 in Appendix N). However, our approach preserves diverse churches in adapted samples and adapts them to target domains. Similar samples can be found in Fig. 7 of CDC, it tends to produce samples of haunted houses even if the source model is pre-trained on cars or horses. AdAM and RICK work on source/target domains with different categories of subjects and get samples of cat faces using the source model pre-trained on FFHQ (human faces).
>
> 2. As illustrated above, our approach shares the same target with CDC and EWC when the source and target domains share the same kind of subjects.
>
> 3. As illustrated in the paper, our approach can be combined with DreamBooth to realize both domain-driven and subject-driven tasks at the same time. Detailed methods and more experiments are provided in Appendix G. It can be seen as an extensive application of the proposed domain-driven method. As for the example, our approach preserves the key features of the target dog, e.g., the same eyes, noses, and ears, and adapts it to the ink painting style.
>
> 2.$\textbf{Task Necessity}$:
>
> We have provided the necessity of this task from two aspects: the ambiguities in natural language and the out-of-distributions effect. Recent works from Google tackle similar tasks as this paper using other text-to-image models [1-2] instead of diffusion models and depending on human feedback. Besides, we have provided an example in the rebuttal above. Given 10-shot landscape drawings, we cannot provide an accurate description with text prompts for Stable Diffusion to synthesize the same style. For another example, in Fig.7 of the revised manuscript, we cannot find a hard text prompt directly for comparison as well. We just use some examples from Van Gogh as examples. Besides, there also exist different styles in Van Gogh's different works. Our approach is designed to learn the style from given samples directly. Moreover, taking DreamBooth as an example, they synthesize results of a Corgi dog given several examples in their paper. However, we cannot say that the subject-driven task is unnecessary since we can directly ask the Stable Diffusion to synthesize samples with a hard text prompt "a Corgi dog" instead of ''a [V] dog".
>
> [1] StyleDrop: Text-to-Image Generation in Any Style, arxiv 2306.00983
>
> [2] Learning disentangled prompts for compositional image synthesis.arXiv:2306.00763
>
> 3.$\textbf{Results from AdAM and RICK}$:
>
> Firstly, we directly run the official implementation code with the provided hyperparameters to make sure the reproduced results are correct. We fix the noise inputs for different GAN-based methods for fair comparison and provide visualized results in Fig. 5 and additional results in Appendix N. The FID results are directly copied from their paper. As for the Intra-LPIPS results, we synthesize images with fixed noise inputs for fair comparison as well. As illustrated in the revised paper, AdAM and RICK still produce low-quality blurs and deformations of human faces, which can be found in Fig. 25 in its supplementary as well. We have added an direct comparison between our results and the visualized samples provided in the publications of AdAM and RICK in Fig.25 of our supplementary to show the advantages of our approach. Our approach achieves apparently better visual effects with high-quality details and avoids unnatural blurs or artifacts which can be found in GAN-based methods, including AdAM and RICK. In our opinion, the visual effects should be the first consideration for generative tasks. The metrics aren’t always super well-aligned with human perception of quality. Besides, the proposed approach is qualified for domain-driven tasks which existing GAN-based methods cannot handle, e.g., LSUN Church $\rightarrow$ Landscape drawings.
>
> Looking forward to your responses. Thanks for your precious time.

---

### Official Review · Reviewer_Qxcc · 2023-11-01

**Soundness:** 2 fair
**Presentation:** 2 fair
**Contribution:** 2 fair
**Rating:** 3
**Confidence:** 4

**Summary:**

The paper proposes a method for domain tuning without overfitting, using approximately 10 chapters of limited data. To achieve this, the paper employs relative distances preservation and high-frequency reconstruction loss.

**Strengths:**

The results seem good without overfitting while they tune the entire set of parameters instead of LoRA. Additionally, it makes sense to maintain the pairwise distance of previously generated images.

**Weaknesses:**

The paper focuses on comparisons with GAN-based approaches and lacks a comparison with LoRA, which is a more commonly used method. Also, there should be more ablation studies such as showing effects of each loss term.

**Questions:**

Is there an improvement in performance compared to the simpler LoRA method? It would be interesting to see the results for ablation studies such as with and without high-frequency reconstruction loss and pairwise similarity loss.

---

> ### Author Response · Authors · 2023-11-15
> **Responses from authors**
>
> Thanks for your precious time for reviews. Here we provide corresponding responses and results to cover your concerns.
>
> 1.$\textbf{LoRA}$: We provide our approach on text-to-image generation with modern few-shot fine-tuning methods based on stable diffusion like textual inversion and dreambooth. We have added additional comparison with LoRA using qualitative (Fig. 36 in Appendix N) and quantitative (Table 6 & 7 in Appendix D) results in the revised manuscript. Similar to DreamBooth, it’s also hard for LoRA to achieve realistic results in domain-driven generation. Thanks for your advice.
>
> 2.$\textbf{Ablations}$: We provided ablation analysis of our approach in Appendix E. Following the advice from Reviewer 4uc5, we have moved this part to the main paper in the revised manuscript. More detailed qualitative and quantitative ablation analysis of each term is provided in Appendix E.
>
> Please let us know if our responses solve your concerns. If you still have any unclear parts about our work, please let us know as well. Thanks for your review.

---

> ### Author Response · Authors · 2023-11-22
>
> Dear reviewer Qxcc,
>
> We thank you for your valuable reviews. We have added experiments of LoRA to show the improvement of our approach in domain-driven tasks. Besides, detailed ablation analysis have been added to the main paper as well. Today is the last day for discussion between authors and reviewers. We are looking forward to your reply.
>
> Thanks for your review!

---

### Author Response · Authors · 2023-11-15
**Thanks for all valuable reviews here**

Thank all reviewers for your valuable reviews. We have revised our paper to give a clearer definition of our task (following reviews from reviewer mvse), revised the experiments and added additional experiments (following reviews from Qxcc and mvse), added detailed explanation of our approach compared with prior works (following reviews form mvse and w5yc) and adjust the whole paper structure and revise the writing (following reviews from mvse and 4uc5). Constrained by the file size limit, we still put the appendix in the supplementary file. We have provided responses to all the concerns proposed by reviewers. We are welcome to further discussion and more suggestions on improving our work. Thanks for your precious time in reviewing our paper!

---

### Author Response · Authors · 2023-11-19

Dear reviewers,

We have provided the rebuttal for your reviews. Could you please look at our responses and the revised paper (revised parts highlighted in blue)? Please let us know what concerns are still not solved or if you still have any unclear parts about our work. We are looking forward to more discussion to improve this paper.

Thanks for your precious time!

---

> ### Author Response · Authors · 2023-11-21
> **About Two Days Left for Discussion**
>
> Dear all reviewers,
>
> There are about two days left for discussion. We have provided the rebuttal for your reviews. Could you please look at our responses and let us know what concerns are still not solved or if you still have any unclear parts about our work? We are looking forward to your responses.
>
> Thanks for your review!

---

> > ### Author Response · Authors · 2023-11-22
> > **One Day Left for Discussion**
> >
> > Dear all reviewers,
> >
> > There is only one day left for discussion. Could you please look at our responses and let us know what concerns are still not solved or if you still have any unclear parts about our work? We are looking forward to your responses sincerely.
> >
> > Thanks for your review!